# Short-Process Preparation of High-Purity V$_2$O$_5$ from Shale Acid Leaching Solution via Chlorination

Dou Huang [1,2,3,4] , Jing Huang [1,2,3,4,*], Yimin Zhang [1,2,3,4,*], Yong Fan [1,2,3,4,*] and Pengcheng Hu [1,2,3,4]

1 School of Resource and Environmental Engineering, Wuhan University of Science and Technology, Wuhan 430081, China; hd970301@126.com (D.H.); hpcmy126@126.com (P.H.)
2 State Environmental Protection Key Laboratory of Mineral Metallurgical Resources Utilization and Pollution Control, Wuhan University of Science and Technology, Wuhan 430081, China
3 Collaborative Innovation Center of Strategic Vanadium Resources Utilization, Wuhan University of Science and Technology, Wuhan 430081, China
4 Hubei Provincial Engineering Technology Research Center of High Efficient Cleaning Utilization for Shale Vanadium Resource, Wuhan University of Science and Technology, Wuhan 430081, China
* Correspondence: crystal208@126.com (J.H.); zhangyimin@wust.edu.cn (Y.Z.); fanyong@wust.edu.cn (Y.F.)

**Abstract:** The conventional V$_2$O$_5$ preparation processes include ion exchange, chemical precipitation, solvent extraction, and other processes. Given the long process and complex operation nature of traditional V$_2$O$_5$ production methods, we herein developed a short-process, low-temperature, and convenient operation method of isolating vanadium (in the form of V$_2$O$_5$) from shale acid leaching solution. The acid leaching solution was oxidized with NaClO$_3$ and pH-adjusted with NaOH to form a vanadium-containing precipitate, which was mixed with AlCl$_3$ (V:AlCl$_3$ = 1:5, mol/mol) and roasted for 120 min at 170 °C to afford vanadium oxytrichloride (VOCl$_3$) with a purity of 99.59%. In addition, the vanadium-containing precipitate was mixed with AlCl$_3$ and NaCl (V:AlCl$_3$:NaCl = 3:12:8, mol/mol/mol) and roasted for 120 min at 170 °C to afford VOCl$_3$ with a purity of 99.94%. VOCl$_3$ (purity of 99.94%) was dissolved in ultrapure water, and the solution (32 g$_{vanadium}$/L) was treated with NH$_3$·H$_2$O (NH$_3$:V = 1.34, mol/mol) at 50 °C for 120 min. The obtained precipitate (vanadium precipitation rate = 99.28%) was roasted at 550 °C for 3 h to afford high-purity vanadium pentoxide (V$_2$O$_5$) with a purity of 99.86%. Compared with the traditional hydrometallurgical method of V$_2$O$_5$ preparation, our method avoided solvent extraction and other undesired processes and the overall process flow is greatly shortened, thus having high practical value.

**Keywords:** chlorination; high-purity vanadium pentoxide; short-process; aluminum chloride; vanadium oxytrichloride



## 1. Introduction

The application scope of vanadium has expanded beyond iron-/steelmaking, military, and medical industries [1–4] to include functional materials (e.g., nanocomposites), high-performance alloys, and all-vanadium liquid-flow batteries [5–13]. In particular, the abovementioned batteries represent a new high-efficiency energy storage technology and energy-development direction, which highlights the strategic importance of vanadium resources [14–16]. The steel industry is a major area of interest for vanadium products. Vanadium has greatly helped to improve the hardness, toughness, wear resistance, and high temperature-resistance of steel alloys products. With the continuous development and expansion of new fields, such as vanadium-based materials, vanadium nitrogen alloys, vanadium electrode products, high-purity vanadium-containing materials, and all vanadium liquid flow batteries, the market demand for high-quality vanadium-containing products will continue to increase. This rapid expansion places high requirements on the quality and quantity of vanadium compounds, of which V$_2$O$_5$ is the most stable and

widely used [17,18]. Therefore, much attention has been drawn to the facile and efficient production of high-purity $V_2O_5$.

$V_2O_5$ is mainly produced through hydrometallurgical and chlorination methods, among which the most common method of separating and purifying metallurgical-grade vanadium products is through hydrometallurgical methods to achieve the purification and enrichment of vanadium [19,20]. However, all hydrometallurgical processes widely used for vanadium recovery, e.g., ion exchange, solvent extraction, adsorption, and precipitation [21], exhibit certain drawbacks. For example, emulsification and flocculation during solvent extraction, rapid extractant loss, organic phase loss through entrainment, re-extraction difficulty, and impurity transfer compromise product quality and complicate organic-phase purification. The ion exchange method has a long cycle time and high salt consumption and generates excessive regeneration waste streams, and the presence of organic substances can contaminate ion exchange resins and discharge a large amount of salty wastewater [22–24], which can easily cause the corrosion of pipelines. In addition, when there are multiple ions in the solution, different resins need to be selected for different ions, resulting in poor universality. The use of chemical precipitation requires the introduction of large amounts of chemical agents, resulting in the secondary pollution of precipitated waste residues.

Unlike their hydrometallurgical counterparts, chlorination-based processes offered the benefits of simplicity, low pollution, low cost, and high selectivity [25–32], and the preparation of $V_2O_5$ from vanadium-containing raw materials with chlorinating agents is therefore drawing much attention [33]. Chlorination is typically performed using $Cl_2$, and the resulting $VOCl_3$ (purity $\geq$ 99.9%) reacts with $NH_3 \cdot H_2O$ in an aqueous medium to afford precipitates that are subsequently converted into $V_2O_5$ [34–36]. Although this method is time-efficient, it requires the handling of the highly corrosive and toxic $Cl_2$ and $VOCl_3$, thus necessitating the use of highly corrosion-resistant industrial equipment and strict safety protocols. In addition, the use of high roasting temperatures increases energy consumption and, hence, production costs. Zheng et al. [37] used $FeCl_X$ as a chlorination agent to extract vanadium from vanadium-bearing titanium magnetite. Under a roasting temperature of 900~1300 K and oxygen atmosphere, the extraction rate of vanadium increases with the increase in temperature and then decreases with the increase in temperature. Using $FeCl_3$ as a chlorinating agent and holding at 1100 K for 2 h, the extraction rate of vanadium can reach 32%. Du et al. [38] extracted 96.36% V and 4.23% Ti from tailings containing 10% petroleum coke via chlorination roasting for 1 h at 800 °C with a chlorine pressure fraction of $[P(Cl_2)/P(Cl_2 + N_2)] = 0.5$. Further purification of the collected chlorinated products resulted in the production of $VOCl_3$ with a purity higher than 99.99%. Wu et al. [27] proposed a method of recovering vanadium from carbonaceous gold ores based on the use of NaCl as a chlorinating agent to separate vanadium, gold, zinc, and iron. However, the temperature of the suggested chlorination reaction is as high as 800 °C. Further, we found that $VOCl_3$ can be prepared below 200 °C when $AlCl_3$ is used as a chlorinating agent. Jiang et al. [1] used anhydrous aluminum chloride and sodium chloride to purify industrial-grade $V_2O_5$ with a purity of 96% at low temperatures of 170 °C, and the high-purity $V_2O_5$ with a purity of at least 99.97% was obtained. The novel method of $V_2O_5$ preparation is highly selective for vanadium. Therefore, using a chlorination method to prepare high-purity $V_2O_5$ has received great attention.

Herein, based on the abovementioned previous works, we developed a short-process method of preparing $V_2O_5$ from shale acid leaching solution. Based on the difference in boiling points of $VOCl_3$ and $FeCl_3$, $MgCl_2$, NaCl, KCl, $CrCl_3$, $MoCl_6$ and $NiCl_2$, through the use of chlorinated metallurgical methods, a vanadium precipitate was obtained by precipitating vanadium from acid leaching solution; the vanadium precipitate was roasted with $AlCl_3$ at a low temperature, and then, a high-purity $VOCl_3$ product was obtained, which was then hydrolyzed through treatment with $NH_3 \cdot H_2O$, and the precipitate was roasted to obtain high-purity $V_2O_5$. Unlike conventional chlorination techniques, our method avoided the ion exchange, chemical precipitation, solvent extraction, and other undesired processes, and the overall preparation process of $V_2O_5$ has been shortened. The

short preparation process of $V_2O_5$ in this study also avoids the use of the toxic and corrosive $Cl_2$ and does not require excessively high temperatures, thus holding great promise for cost-effective and facile vanadium recovery.

## 2. Materials and Methods

### 2.1. Materials

The acid leaching solution used in this study originated from Xi'an, Shanxi, China. Table 1 lists the concentrations of major elements in the shale acid leaching solution used as the raw material. Other reagents were chemically pure and did not require further purification. Nitrogen (99.999%, Wuhan NRD, China) was used as a protective gas, and ultrapure water was used throughout the experiments.

**Table 1.** Concentrations of major elements in the shale acid leaching solution (g/L).

| Element | V | Na | K | Mg | Fe | Al | P | Ni | Mo | Cr | S |
|---|---|---|---|---|---|---|---|---|---|---|---|
| Concentration | 2.32 | 0.69 | 0.81 | 14.02 | 3.76 | 2.11 | 0.58 | 1.74 | 0.02 | 0.23 | 51.24 |

### 2.2. Procedure for Short-Process Preparation of $V_2O_5$

The short-process preparation of $V_2O_5$ includes three steps; the first step is vanadium precipitation from the shale acid leaching solution, and the precipitate is obtained. The second step is precipitate chlorination (using nitrogen for protection); vanadium and impurities of the precipitate are separated in one step via chlorination, and the high-purity $VOCl_3$ is obtained. The third step is ammonolysis of the $VOCl_3$ to obtain $NH_4VO_3$, which is calcined to obtain $V_2O_5$. The diluted sulfuric acid solution is used to absorb the ammonia gas generated during the calcination process. Figure 1 represents the flow chart of this process.

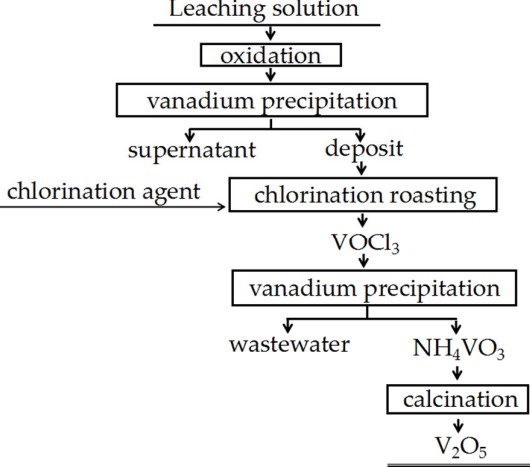

**Figure 1.** Flow chart of short-process preparation of $V_2O_5$.

#### 2.2.1. Precipitation of Vanadium from Acid Leaching Solution

The acid leaching solution was oxidized by adding $NaClO_3$ (2.8 g $NaClO_3$ was added to 500 mL of acid leaching solution), pH-adjusted to 5.5 using NaOH, and stirred at 30 °C for 150 min. The precipitate was isolated via filtration and dried at 60 °C for 3 h.

#### 2.2.2. Chlorination of Precipitate Isolated from Leaching Solution

Before conducting chlorination experiments, nitrogen gas needs to be continuously introduced into the reaction device to eliminate excess water vapor. The precipitate isolated from the leaching solution was mixed with the chlorinating agent and roasted in the three-necked flask at a set roasting temperatures and times. The generated gaseous $VOCl_3$ was condensed into a liquid and collected in a flask. The tail gas treatment device equipped

with NaOH solution was used to absorb the uncondensed gaseous $VOCl_3$ and other tail gases in the chlorination reaction. The diagram of the chlorination reaction experimental device is shown in Figure 2.

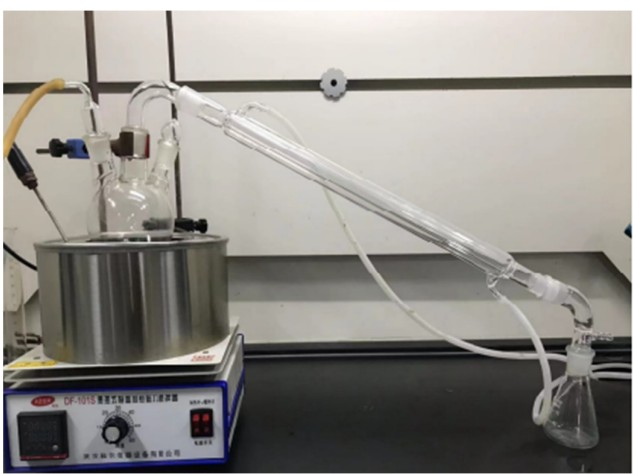

**Figure 2.** The experimental device of the chlorination reaction.

The vanadium extraction rate ($\omega_v$) was calculated as follows:

$$\omega_v = \frac{m_1 \times 50.942}{m_0 \omega_0 \times 173.299} \times 100\%, \tag{1}$$

where $m_0$ is the mass of the vanadium-containing precipitate (g), $\omega_0$ is the mass fraction of vanadium in the precipitate, $m_1$ is the mass of $VOCl_3$ (g), 50.942 is the molar mass of V (g/mol), and 173.299 is the molar mass of $VOCl_3$ (g/mol).

### 2.2.3. Ammonolysis of $VOCl_3$

$VOCl_3$ is very unstable, generating red smoke upon contact with water or water vapor. Herein, $VOCl_3$ was dissolved in ultrapure water, and the solution was treated with $NH_3 \cdot H_2O$ at different $NH_3$:V molar ratios. The resulting precipitate was roasted at 550 °C for 3 h to afford $V_2O_5$.

### 2.3. Characterization of $VOCl_3$ and $V_2O_5$

The concentrations of major elements in the shale acid leaching solution, the composition content of the precipitate isolated from the acid leaching solution, the impurity content in the $VOCl_3$, and the high-purity $V_2O_5$ product were determined via inductively coupled plasma optical emission spectroscopy (ICP-OES; model 730, Agilent Co., Ltd., Shanghai, China). The compositions of $V_2O_5$ and the chlorination residue were determined via X-ray diffraction (XRD) (Smart-Lab SE, Rigaku Co., Ltd., Tokyo, Japan). The operation conditions for XRD were as follows: light tube voltage, 40 Kv; light tube current, 40 Ma; scanning angle range, 5°–90°; scanning speed, 15.6°/min. Moreover, micro-morphologies and elemental distributions were determined via scanning electron microscopy–energy-dispersive X-ray spectroscopy (SEM–EDS) (JSM-IT300, JEOL Co., Ltd., Tokyo, Japan). The operation conditions for SEM–EDS were as follows: acceleration voltage, 20–30 Kv; the mode was switched to high current one when analyzing samples. The pH of the solution was measured using a pH meter (PHS-3C Shanghai INESA Scientific Instrument Co., Ltd., Shanghai, China).

## 3. Results and Discussion

### 3.1. Analysis of Precipitate Isolated from Acid Leaching Solution

Table 2 represents the composition of the resulting precipitate, while Figures 3 and 4 represent the corresponding XRD pattern and SEM–EDS data, respectively.

**Table 2.** Composition (wt%) of the precipitate isolated from the acid leaching solution.

| Component | $V_2O_5$ | $Fe_2O_3$ | MgO | $Al_2O_3$ | $Na_2O$ | $Cr_2O_3$ | $P_2O_5$ | NiO | $MoO_3$ | $K_2O$ | $SO_3$ |
|---|---|---|---|---|---|---|---|---|---|---|---|
| **Content** | 7.52 | 3.81 | 14.10 | 12.88 | 17.45 | 0.21 | 0.68 | 0.43 | 0.15 | 1.76 | 36.15 |

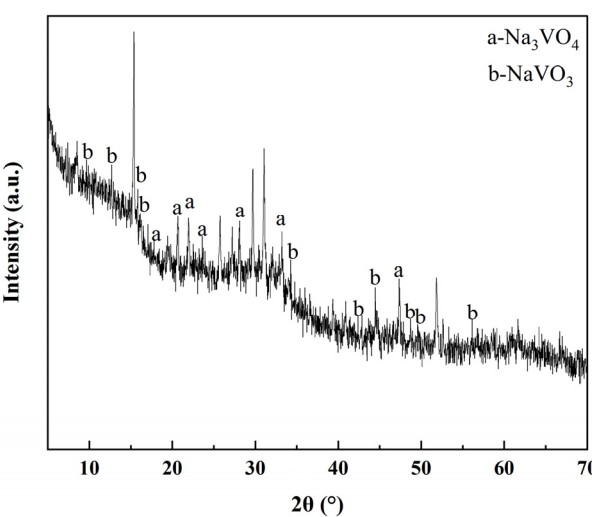

**Figure 3.** XRD pattern of the precipitate isolated from the acid leaching solution.

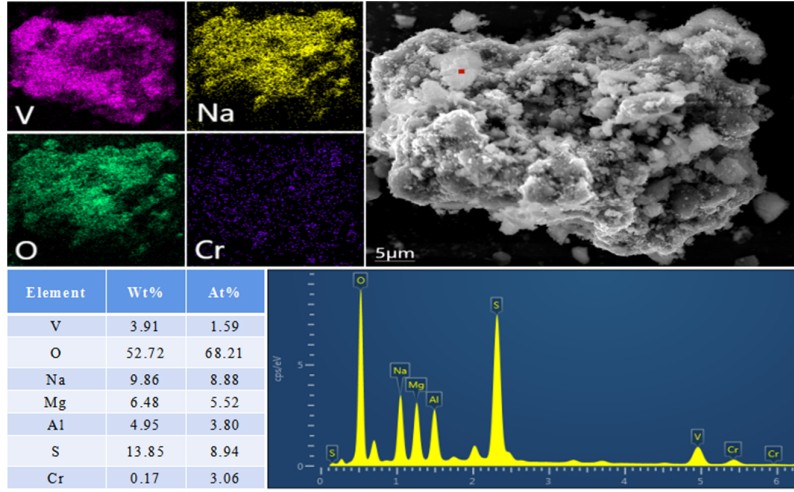

**Figure 4.** SEM–EDS characterization of the precipitate isolated from the acid leaching solution.

Figure 3 shows that the diffraction peaks of $Na_3VO_4$ and $NaVO_3$ were present in the XRD patterns of the precipitate isolated from the acid leaching solution. Figure 4 shows that in addition to vanadium, the precipitate contained sodium and oxygen, which is consistent with the results of the XRD analysis. The precipitate contained a significant amount of sodium; the reason is that the acid leaching solution introduces a large amount of sodium when the pH is adjusted with NaOH. Due to the high concentrations of magnesium, aluminum, and sulfur in the acid leaching solution, magnesium, aluminum, and sulfur are introduced into the precipitate during the vanadium precipitation process.

### 3.2. Thermodynamic Analysis of Chlorination Reaction

The precipitate isolated from the acid leaching solution was used as raw material, and $AlCl_3$ was used as the chlorination agent. Based on the composition of the precipitate

isolated from the acid leaching solution, the following reactions (R1)~(R10) may occur in the chlorinated system:

$$Na_3VO_4 + 2AlCl_3 = VOCl_3 + Al_2O_3 + 3NaCl \tag{R1}$$

$$3NaVO_3 + 4AlCl_3 = 3VOCl_3 + 2Al_2O_3 + 3NaCl \tag{R2}$$

$$3Na_2O + 2AlCl_3 = Al_2O_3 + 6NaCl \tag{R3}$$

$$Fe_2O_3 + 2AlCl_3 = Al_2O_3 + 2FeCl_3 \tag{R4}$$

$$3MgO + 2AlCl_3 = Al_2O_3 + 3MgCl_2 \tag{R5}$$

$$Cr_2O_3 + 2AlCl_3 = Al_2O_3 + 2CrCl_3 \tag{R6}$$

$$3P_2O_5 + 10AlCl_3 = 5Al_2O_3 + 6PCl_5 \tag{R7}$$

$$3NiO + 2AlCl_3 = Al_2O_3 + 3NiCl_2 \tag{R8}$$

$$MoO_3 + 2AlCl_3 = Al_2O_3 + MoCl_6 \tag{R9}$$

$$3K_2O + 2AlCl_3 = Al_2O_3 + 6KCl \tag{R10}$$

The Gibbs free energy changes of the above reactions, (R1)~(R10), at different temperatures were calculated based on corresponding thermodynamic data from the HSC 6.0 software package, as shown in Figure 5.

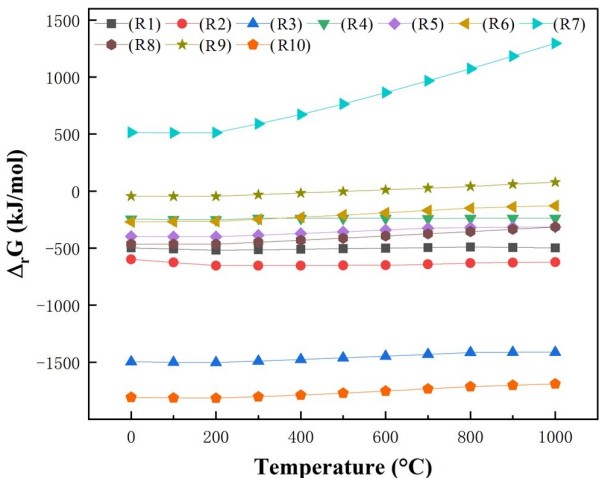

**Figure 5.** Gibbs free energy changes of related reactions at different temperatures.

Figure 5 shows that AlCl$_3$ was used as a chlorination agent, and Gibbs free energy changes were negative below 1000 °C for the chlorination reaction of Na$_3$VO$_4$ and NaVO$_3$, which indicates that VOCl$_3$ can be formed in chlorinated systems. Gibbs free energy changes were positive below 1000 °C for the chlorination reaction of P$_2$O$_5$, which indicates that the chlorination reaction of P$_2$O$_5$ is not favorable. Even though Gibbs free energy changes were negative below 500 °C for the chlorination reaction of Na$_2$O, Fe$_2$O$_3$, MgO, Cr$_2$O$_3$, NiO, MoO$_3$, and K$_2$O, according to Table 3, we know that FeCl$_3$, MgCl$_2$, NaCl,

$MoCl_6$, $NiCl_2$, $CrCl_3$, and KCl were non-volatile below 200 °C. Therefore, we speculated that when $AlCl_3$ functioned with the vanadium-containing precipitate isolated from the acid leaching solution, the high-purity $VOCl_3$ could be prepared.

**Table 3.** Physical and chemical properties of chloride.

| Chloride | Melting Point/°C | Boiling Point/°C |
|---|---|---|
| $VOCl_3$ | −77 | 126 |
| $FeCl_3$ | 306 | 316 |
| $MgCl_2$ | 714 | 1412 |
| $AlCl_3$ | 194 | 178 |
| NaCl | 801 | 1465 |
| $MoCl_6$ | 249 | 352 |
| $NiCl_2$ | 1001 | 973 |
| $CrCl_3$ | 1152 | 1300 |
| $PCl_5$ | 180 | 375 |
| KCl | 773 | 1500 |

### 3.3. Effects of Chlorination Parameters

#### 3.3.1. Temperature

The effects of the chlorination temperature (100–220 °C) were explored under a protective atmosphere of nitrogen at a reaction time of 120 min and a $V:AlCl_3$ molar ratio of 1:4 (Figure 6).

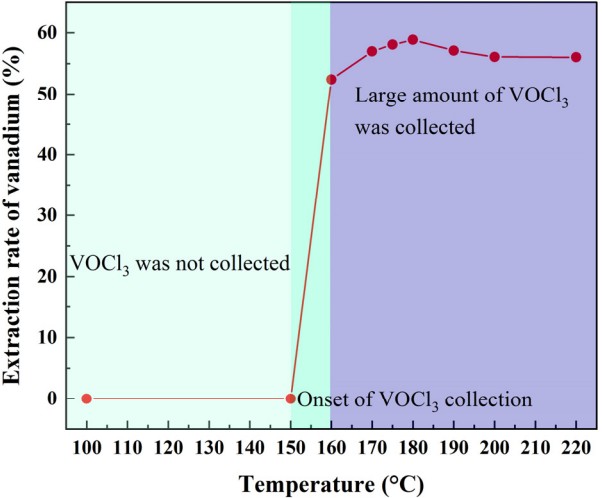

**Figure 6.** Effects of chlorination temperature on the extraction rate of vanadium.

Figure 6 shows that $VOCl_3$ collection started at 150 °C and that the extraction rate increased with increasing temperature and reached a maximum at 180 °C. After 180 °C, the extraction rate decreased with increasing temperature and plateaued after 200 °C. The decrease in the vanadium extraction rate is due to the boiling point of $AlCl_3$ being 178.8 °C, and when the chlorination reaction temperature is above 180 °C, $AlCl_3$ is heavily sublimated, resulting in a less efficient chlorination reaction.

#### 3.3.2. Dosage of $AlCl_3$

The effects of the $AlCl_3$ dosage ($V:AlCl_3$ = 1:1, 1:2, 1:3, 1:4, and 1:5 mol/mol) on the vanadium extraction rate and $VOCl_3$ purity were examined at 170 °C using a reaction time of 120 min and nitrogen as the carrier and protective gas (Figure 7 and Table 4).

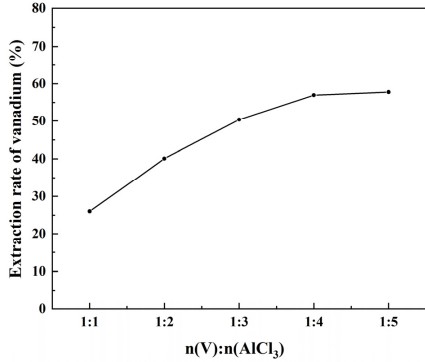

**Figure 7.** Effects of AlCl$_3$ dosage on the vanadium extraction rate.

**Table 4.** Effects of AlCl$_3$ dosage on the purity of VOCl$_3$.

| N(V):n(AlCl$_3$) | Impurity Content (%) | | | | | | | | | VOCl$_3$ Purity (%) |
|---|---|---|---|---|---|---|---|---|---|---|
| | **Na** | **Fe** | **Mg** | **Al** | **Ni** | **Mo** | **Cr** | **P** | **K** | |
| 1:1 | 0.1157 | 0.1134 | 0.1016 | 0.1970 | 0.1061 | 0.0517 | 0.1110 | 0.0515 | 0.1012 | 99.05% |
| 1:2 | 0.1166 | 0.1032 | 0.1028 | 0.1365 | 0.1075 | 0.0250 | 0.1032 | 0.0222 | 0.0230 | 99.26% |
| 1:3 | 0.1078 | 0.1009 | 0.1021 | 0.1083 | 0.1063 | 0.0508 | 0.1004 | 0.0024 | 0.0515 | 99.27% |
| 1:4 | 0.1053 | 0.1055 | 0.1044 | 0.1040 | 0.0507 | 0.0503 | 0.0505 | 0.0193 | 0.0505 | 99.38% |
| 1:5 | 0.0531 | 0.0531 | 0.0557 | 0.1065 | 0.0502 | 0.0201 | 0.0502 | 0.0101 | 0.0101 | 99.59% |

With an increasing AlCl$_3$ dosage, the vanadium extraction rate increased, while the purity of VOCl$_3$ remained above 99%. Thus, considering the vanadium extraction rate, VOCl$_3$ purity, and cost, we identified the optimum V:AlCl$_3$ molar ratio as 1:5, which corresponded to a VOCl$_3$ purity of 99.59%.

3.3.3. Effect of Temperature on Chlorination in the Presence of NaCl

Based on FactSage-7.2 software package, with an AlCl$_3$:NaCl molar ratio is 3:2, the chlorination temperature is above 156.70 °C and AlCl$_3$ and NaCl can form a molten salt liquid (NaAlCl$_4$). Further, the effects of temperature on chlorination in the presence of NaCl were explored under a protective atmosphere of nitrogen at a reaction time of 120 min, a V:AlCl$_3$ molar ratio of 1:4, and an AlCl$_3$:NaCl molar ratio of 3:2.

Figure 8 shows that VOCl$_3$ collection started at 150 °C and that the extraction rate increased with an increasing temperature and plateaued after 160 °C. AlCl$_3$ and NaCl were used as the chlorinating agent, and there was no AlCl$_3$ sublimation; the reason is that the boiling point of NaAlCl$_4$ is much higher than the chlorination reaction temperature. Therefore, 170 °C was chosen as the optimal temperature.

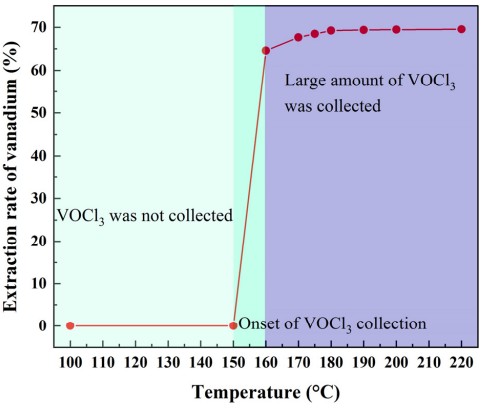

**Figure 8.** Effects of temperature on the vanadium extraction rate in the presence of NaCl.

### 3.3.4. Effect of AlCl$_3$ and NaCl Dosage on Chlorination

The effects of the AlCl$_3$ and NaCl dosage (V:AlCl$_3$:NaCl = 3:3:2, 3:6:4, 3:9:6, 3:12:8, and 3:15:10 mol/mol) on the vanadium extraction rate and VOCl$_3$ purity were examined at 170 °C using a reaction time of 120 min and nitrogen as the carrier and protective gas (Figure 9 and Table 5).

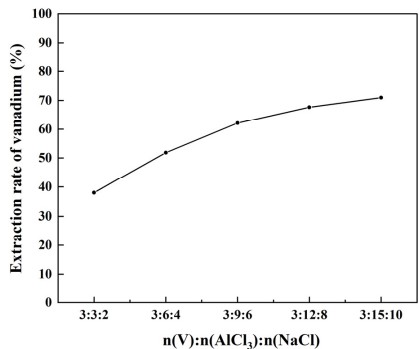

**Figure 9.** Effects of AlCl$_3$ and NaCl dosage on the vanadium extraction rate.

**Table 5.** Effects of AlCl$_3$ and NaCl dosage on the purity of VOCl$_3$.

| N(V):n(AlCl$_3$):n(NaCl) | Impurity Content (%) | | | | | | | | | VOCl$_3$ Purity (%) |
|---|---|---|---|---|---|---|---|---|---|---|
| | **Na** | **Fe** | **Mg** | **Al** | **Ni** | **Mo** | **Cr** | **P** | **K** | |
| 3:3:2 | 0.0157 | 0.0134 | 0.0016 | 0.0470 | 0.0061 | 0.0017 | 0.0110 | 0.0015 | 0.0012 | 99.91% |
| 3:6:4 | 0.0166 | 0.0032 | 0.0028 | 0.0465 | 0.0075 | 0.0030 | 0.0032 | 0.0002 | 0.0030 | 99.91% |
| 3:9:6 | 0.0078 | 0.0009 | 0.0021 | 0.0483 | 0.0063 | 0.0008 | 0.0004 | 0.0004 | 0.0005 | 99.93% |
| 3:12:8 | 0.0053 | 0.0055 | 0.0044 | 0.0440 | 0.0007 | 0.0003 | 0.0005 | 0.0003 | 0.0005 | 99.94% |
| 3:15:10 | 0.0031 | 0.0031 | 0.0057 | 0.0465 | 0.0002 | 0.0001 | 0.0002 | 0.0001 | 0.0001 | 99.95% |

Molten salt liquid (NaAlCl$_4$) can improve the vanadium extraction rate and purity of VOCl$_3$. With the addition of NaCl, the vanadium extraction rate increased, while the purity of VOCl$_3$ remained above 99.9%. Thus, considering the vanadium extraction rate, VOCl$_3$ purity, and cost, we identified the optimum V:AlCl$_3$:NaCl molar ratio as 3:12:8, which corresponded to a VOCl$_3$ purity of 99.94%.

### 3.4. Analysis of Chlorination Residue

After the chlorination reaction was completed, fibrous substances were found in the chlorination residue. Afterwards, we extracted the fibrous substances from the chlorination residue for analysis. Figure 10 shows that aluminum, oxygen, and chlorine had obvious interactions in the fibrous substances, which had an Al:O:Cl molar ratio close to 1:1:1 according to SEM–EDS analysis.

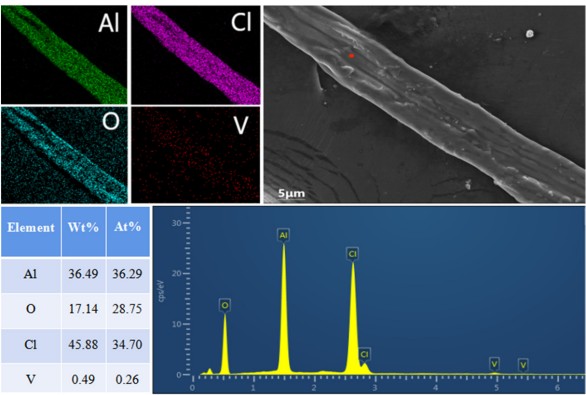

**Figure 10.** SEM–EDS characterization of fibrous substances formed during chlorination.

Figure 11 represents the XRD pattern of the chlorination residue and shows that the diffraction peaks of $NaAlCl_4$ and AlOCl appeared in the chlorination residue, which is consistent with the SEM–EDS analysis of fibrous substances and the fact that $AlCl_3$ and NaCl can form a molten salt liquid ($NaAlCl_4$). Thus, we concluded that these fibrous substances corresponded to AlOCl.

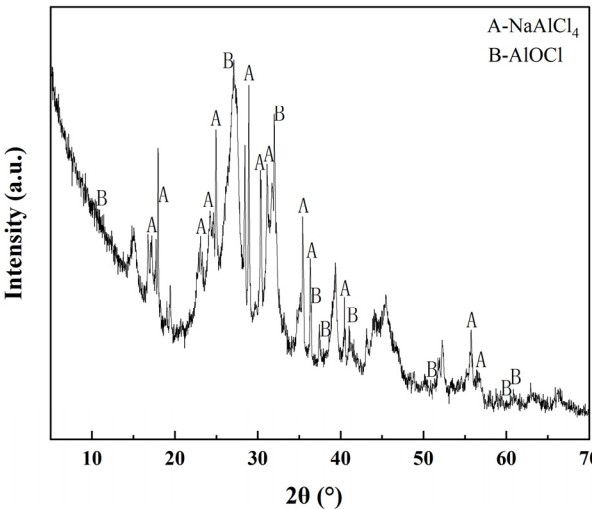

**Figure 11.** XRD pattern of chlorination residue during chlorination.

Based on the above conclusion, regardless of the presence of NaCl in the chlorination system, we speculate that the chlorination reaction mechanism is that the oxygen atoms in $Na_3VO_4$ and $NaVO_3$ first replace the two chlorine atoms in $AlCl_3$. The replaced chlorine atoms chlorinate $Na_3VO_4$ and $NaVO_3$ to generate $VOCl_3$. The chlorine atom in $AlCl_3$ is gradually replaced, resulting in a transition from $AlCl_3$ to AlOCl to $Al_2O_3$. The corresponding chemical equations are as follows:

$$Na_3VO_4 + 3AlCl_3 = 3AlOCl + VOCl_3 + 3NaCl \tag{R11}$$

$$NaVO_3 + 2AlCl_3 = 2AlOCl + VOCl_3 + NaCl \tag{R12}$$

$$6AlOCl + Na_3VO_4 = VOCl_3 + 3NaCl + 3Al_2O_3 \tag{R13}$$

$$4AlOCl + NaVO_3 = VOCl_3 + NaCl + 2Al_2O_3 \tag{R14}$$

*3.5. $V_2O_5$ Preparation from $VOCl_3$ Solution*

3.5.1. Effects of $VOCl_3$ Hydrolysis Parameters

Given the extreme instability of $VOCl_3$ and considering the loss of vanadium and operation feasibility, we dissolved $VOCl_3$ in ultrapure water to prepare a solution with a vanadium concentration of 32 g/L and then treated it with $NH_3 \cdot H_2O$ to afford a precipitate that was subsequently converted to $V_2O_5$.

Figure 12a–c shows the influence of the $NH_3$:V molar ratio, precipitation temperature, and precipitation time on the vanadium precipitation rate. For this investigation, the vanadium precipitation temperature was 20 °C and the vanadium precipitation time was 30 min. With an increasing $NH_3$:V molar ratio, the vanadium precipitation rate increased to a maximum of 96.54% at a ratio of 1.34 and then decreased, which was ascribed to the dissolution of vanadium species in the presence of excess ammonia. Thus, the $NH_3$:V molar ratio of 1.34 was selected as optimal.

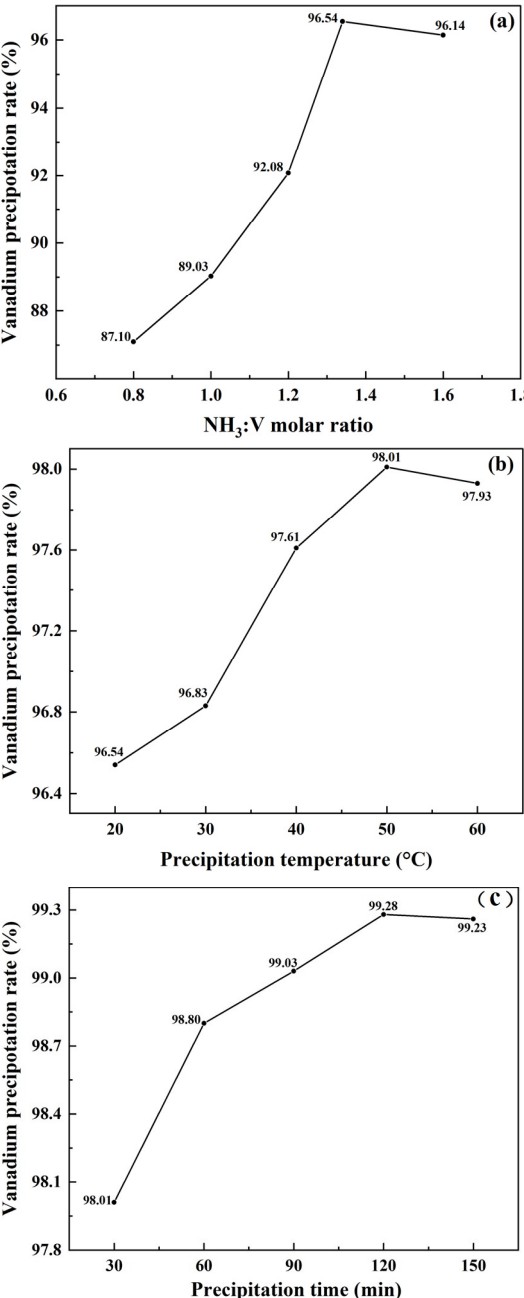

**Figure 12.** Effects of (**a**) the NH$_3$:V molar ratio, (**b**) precipitation temperature, and (**c**) precipitation time on the vanadium precipitation rate.

Furthermore, at the NH$_3$:V molar ratio of 1.34 and the vanadium precipitation time of 30 min, with an increase in temperature, the vanadium precipitation rate increased to a maximum of 98.01% at 50 °C and then decreased because of the solubilization of vanadium species at high temperatures. Therefore, the optimal vanadium precipitation temperature was selected as 50 °C.

At the NH$_3$:V molar ratio of 1.34 and the vanadium precipitation temperature of 50 °C, with an increasing precipitation time, the vanadium precipitation rate increased to a maximum of 99.28% at 120 min and then slightly decreased because of the dissolution of some vanadium species during prolonged exposure to the aqueous solution. Thus, the optimal vanadium precipitation time was selected as 120 min.

### 3.5.2. Characterization of $V_2O_5$

Based on the above optimal conditions, $V_2O_5$ was prepared by roasting the precipitate isolated from the $VOCl_3$ solution at 550 °C for 3 h. The obtained product had a purity of 99.86% and thus met the standard of grade 99 $V_2O_5$ stipulated by YB/T 5304-2011 [39]. Table 6 lists the compositions of standard $V_2O_5$ and that obtained herein, while Figure 13 shows the XRD pattern of the $V_2O_5$ obtained herein and the reference $V_2O_5$ pattern.

**Table 6.** Compositions of the $V_2O_5$ standard and the $V_2O_5$ obtained herein (wt%).

| Constituent | $V_2O_5$ | Si | Fe | P | S | As | $Na_2O + K_2O$ |
|---|---|---|---|---|---|---|---|
| $V_2O_5$ (99% standard *) | >99 | <0.15 | <0.20 | <0.03 | <0.01 | <0.01 | <1.0 |
| $V_2O_5$ obtained herein | 99.86 | 0.0005 | 0.0384 | 0.0004 | 0.0087 | 0.0005 | 0.0315 |

* Standard reference YB/T 5304-2011.

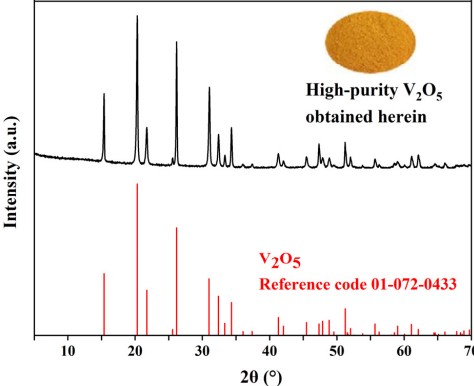

**Figure 13.** XRD pattern and image (inset) of $V_2O_5$ obtained herein.

The XRD pattern of our $V_2O_5$ matched well with that of the $V_2O_5$ standard (standard card code 01-072-0433) and did not contain any impurity peaks.

Figure 14 shows the results of the surface SEM–EDS analysis of the $V_2O_5$ product, revealing that the $V_2O_5$ particles were associated with sodium and magnesium, which indicated a part of sodium and magnesium present in the precipitate isolated from the $VOCl_3$ solution, which was consistent with the high concentrations of sodium and magnesium in the precipitate isolated from the acid leaching solution. Further, sodium and magnesium were confirmed to be the main impurity via point-scan elemental composition analysis. Moreover, carbon was derived from the substrate adhesive.

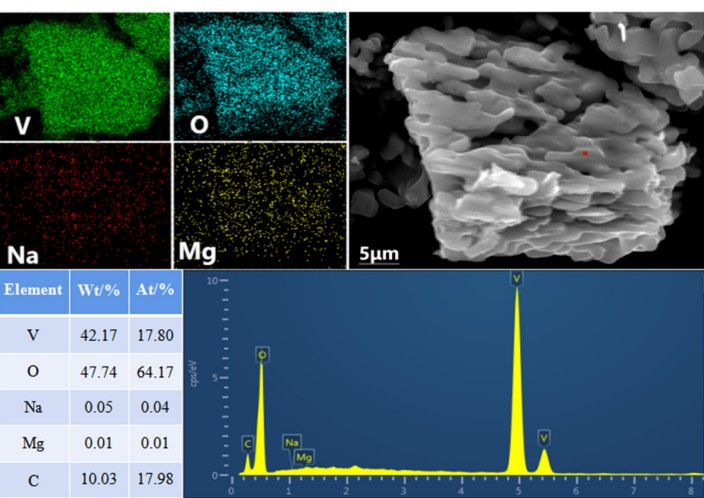

**Figure 14.** SEM–EDS characterization of $V_2O_5$ obtained herein.

### 3.6. Comparison with Traditional Hydrometallurgical Method of $V_2O_5$ Preparation

As shown in Figure 15, compared with the traditional $V_2O_5$ preparation process [40], the short-process preparation of $V_2O_5$ in this study avoided the steps of the reduction of the neutralization solution, solvent extraction of the extraction solution, and backwash extractor of the loaded organic phase. By using the chlorination metallurgy method, the chlorination agent was mixed with the vanadium-containing precipitate from the acid leaching solution and roasted for 120 min at 170 °C to afford $VOCl_3$ with a purity of at least 99.9%, vanadium and impurities were separated in one step, and the overall process flow was shortened. A short process to produce $V_2O_5$ from vanadium shale at a low temperature via chlorination had been achieved. In addition, compared to the traditional $V_2O_5$ preparation process with a vanadium recovery rate of 83.81%, the vanadium recovery rate in this study was only 60.03%, and we will optimize the recovery rate data in the next experimental work. Currently, the focus of this article is to prepare a high-purity $VOCl_3$ intermediate product with a purity of 99.94% from the vanadium precipitate; vanadium and impurities were separated in one step, and then, $V_2O_5$ with a purity of 99.86% was prepared in a short process.

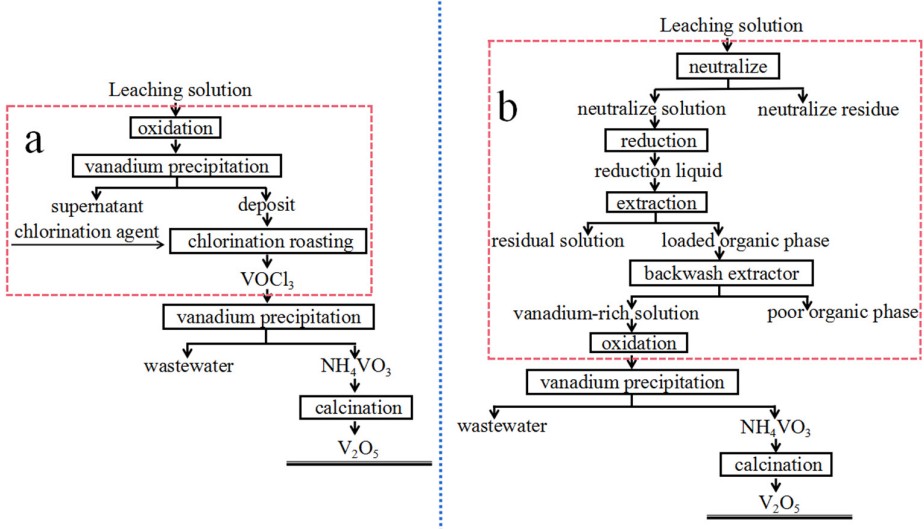

**Figure 15.** $V_2O_5$ preparation process flow chart of (**a**) newly developed and (**b**) traditional methods.

### 4. Conclusions

This study describes a short process of preparing $V_2O_5$ from shale acid leaching solution at a low temperature via precipitation followed by chlorination with $AlCl_3$ at a low temperature (170 °C). The addition of NaCl can improve the vanadium extraction rate and purity of $VOCl_3$. The obtained $VOCl_3$ (purity $\geq$ 99.9%) is dissolved in ultrapure water, the solution is treated with aqueous ammonia, and the precipitate is roasted to afford $V_2O_5$. The optimal process parameters are as follows: chlorination temperature = 170 °C, chlorination time = 120 min, V:AlCl$_3$:NaCl = 3:12:8 (mol/mol/mol) [chlorination], precipitation temperature = 50 °C, precipitation time = 120 min, NH$_3$:V = 1.34 (mol/mol), and vanadium concentration = 32 g/L [precipitation]. The $V_2O_5$ obtained under the optimal conditions had a purity of 99.86%. Compared with the traditional hydrometallurgical process of $V_2O_5$ preparation, our method is characterized by simple operation and a short preparation process.

In addition, we found fibrous substances in the residue of the chlorination reaction. By means of XRD and SEM–EDS analysis of fibrous substances, this was determined to be AlOCl. Based on the formation of AlOCl during the chlorination reaction, the chlorination reaction mechanism was defined as the oxygen atoms in $Na_3VO_4$ and $NaVO_3$ first replacing the two chlorine atoms in $AlCl_3$ and the replaced chlorine atoms chlorinating $Na_3VO_4$ and $NaVO_3$ to generate $VOCl_3$. The chlorine atom in $AlCl_3$ is gradually replaced.

**Author Contributions:** Conceptualization, D.H.; methodology, D.H.; investigation, D.H.; writing—original draft, D.H.; writing—review and editing, D.H., J.H., Y.Z., Y.F. and P.H.; validation, J.H., Y.Z. and P.H.; resources, J.H., Y.Z. and Y.F.; supervision, J.H., Y.Z., Y.F. and P.H.; funding acquisition, J.H., Y.Z. and Y.F. All authors have read and agreed to the published version of the manuscript.

**Funding:** This research was financially supported by the National Natural Science Foundation of China (No. 51974207), the National Key R&D Program of China (2020YFC1909700, 2021YFC2901600), and the Science and Technology Innovation Talent program of Hubei Province (2022EJD002).

**Data Availability Statement:** The data that support the findings of this study are available from the corresponding author (Jing Huang) upon reasonable request.

**Conflicts of Interest:** The authors declare no conflict of interest. The funders had no role in the design of the study; in the collection, analyses, or interpretation of data; in the writing of the manuscript; or in the decision to publish the results.

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
