# Peer review of "Short-Process Preparation of High-Purity V2O5 from Shale Acid Leaching Solution via Chlorination"

_processes, doi:10.3390/pr11041270_

Round 1

Reviewer 1 Report

This manuscript presents an interesting work using oxidation/neutralization-chlorination instead of neutralization-reduction-solvent extraction-oxidation to enrich and purify vanadium from acid leach solution. It is original and interesting, although it can be further improved.

The following are some points for further consideration:

1. The statement in Lines 78-81 does not make sense. Needing a revision.

2. Figure 2: the identification of the major peaks are missing. It is important that all the major peaks are identified. In fact, the XRD pattern looks quite amorphous. There may be some hydrated minerals or others in the precipitates that cannot be identified by XRD.

3. The chemical composition (Table 2) and SEM-EDX analysis do not match. Table 2 shows that V accounts for around 13wt% while the SEM-EDX shows only around 3.9wt% of V. The difference is obvious.

4. The recovery rate through chlorination is around 70%. It needs further improvement. The relatively low recovery rate may reduce the attractions and interests of readers in this work since a lot of energy/chemicals have been consumed during the acid leaching process.

5. An analysis on energy consumption is recommended. The process, low temperature (<100C) oxidation - moderate temperature (around 170C) chlorination - low temperation (<100C) precipitation - high temperature (around 550C) calcination may consumed a lot of energy to raise temperature of water throughout the process. A shorter and simpler process would be more interesting.

Reviewer 2 Report

Review of the article processes-2335251: Short-processes preparation of high-purity V2O5 from shale acid leaching solution by chlorination.

The article under review treats vanadium recovery by short-process that includes oxidation/precipitation-roasting/precipitation-calcination steps from an acid-leaching solution. The studied process presents some advantages in comparison with traditional vanadium recovery processes. The article seems to be well written and the information is well presented, however, some comments should be addressed before its publication:

Comments:

- In the methodology section, the initial volume of the acid-leaching solution treated must be provided.

- Is it possible to provide an efficiency value of the process? Which was the yield of the proposed process? A metallurgical balance of every step that forms the proposed method of extraction must be provided. In the article, the total recovery of vanadium from the initial sample even was not mentioned. For any hydrometallurgical process, this value must be provided.

- In the same sense as the previous comment, which is the efficiency of the vanadium recovery with the traditional process? This value would be useful for an appropriate comparison between processes of extraction.

- In the analysis of the chlorination residue (section 3.4) it was mentioned that in the DRX results the presence of sodium as NaAlCl4 was stated, however in the SEM analysis results sodium was not detected.

- Is there a characterization result for the aminolysis step?

- In my opinion, the FTIR results should be removed from the article since does not provide important information regarding the proposed process.

- The font size in some figures is too small and is difficult to follow, for example, Fig. 12 and EDS spectrums in Figs. 3, 10.

Reviewer 3 Report

This article explores the preparation of high-purity vanadium pentoxide using a chlorination distillation process. This process shows significant advantages over existing processes. The article is highly innovative and has the potential for acceptance. The article needs major revisions in the following sections:

1. The literature review section is incomplete and needs to be summarized again to highlight the advantages and disadvantages of the preparation process of high-purity vanadium pentoxide.

2. The purity detection process of high purity vanadium pentoxide is not clear, and how vanadium and impurity elements are detected is unclear.

3. The description of the experimental process in the article is too simple and requires a detailed procedures.

4. The conclusions of the article need to be summarized again, with emphasis on the discussion of the reaction mechanism.
